# A Novel Study of β1- and β2-Adrenergic Receptors Present on PBMCs, T Cells, Monocytes, and NK Cells by Radioligand Method: Quantitation and Correlations

**DOI:** 10.3390/ijms26167894

**Published:** 2025-08-15

**Authors:** Mihail. M. Peklo, Ekaterina V. Smolyakova, Lyudmila N. Lipatova, Natal’ya M. Kashirina, Yurij S. Skoblov, Natal’ya A. Skoblova, Mihail A. Slinkin, Igor’ N. Rybalkin, Pavel N. Rutkevich, Olga K. Chusovitina, Elena V. Yanushevskaya, Kirill A. Zykov, Tat’yana N. Vlasik

**Affiliations:** 1National Medical Research Center of Cardiology Named After Academician E. I. Chazov, 121552 Moscow, Russia; liliudon3005@yandex.ru (L.N.L.); kashka55@yandex.ru (N.M.K.); mikh.slinkin@gmail.com (M.A.S.); irybalkin@yahoo.com (I.N.R.); p.rutkevich@gmail.com (P.N.R.); tchuso@yandex.ru (O.K.C.); yanushevskaya@yandex.ru (E.V.Y.); kirillaz@inbox.ru (K.A.Z.); tanya.vlasik@gmail.com (T.N.V.); 2Federal State Budgetary Educational Institution of Higher Education “Russian University of Medicine” of the Ministry of Healthcare of the Russian Federation, 127006 Moscow, Russia; smolyakovak@mail.ru; 3Institute of Bioorganic Chemistry Named After Academicians M. M. Shemyakin and Y. A. Ovchinnikov, 117997 Moscow, Russia; uskoblov@gmail.com (Y.S.S.); skobond@gmail.com (N.A.S.)

**Keywords:** β1-adrenoreceptor, β2-adrenoreceptor, T cells, monocytes, NK cells

## Abstract

β-adrenoreceptor (ADRB) ligands are actively used in the therapy of bronchopulmonary and cardiovascular diseases. When using these drugs, it is important to assess changes in ADRB content in different tissues. In most cases, the direct measurement of ADRB content in lung and heart cells is not possible. ADRB2 content in peripheral blood lymphocytes (or mononuclear cells) was shown to correlate with that in myocardial cells. It has been suggested that blood lymphocytes can be used to monitor ADRB content in solid organs. However, the estimation of ADRB1 content in myocardium from its content in peripheral lymphocytes is not possible due to the low content of ADRB1 in lymphocytes. In the present study, we performed simultaneous determination of ADRB1 and ADRB2 both in the total population of PBMCs and in isolated subpopulations of monocytes, T-lymphocytes, and NK-cells from 23 healthy donors using the modified radioligand method. The highest amount of ADRB2 was detected in NK cells, followed by PBMCs, monocytes, and T cells. The content of these receptors in all blood cell subpopulations was significantly correlated with each other, suggesting the possibility of using PBMCs to monitor ADRB2 in solid organs. For the first time, ADRB1 was detected in monocytes and NK cells.

## 1. Introduction

Adrenergic mechanisms play an important role in the pathogenesis of cardiovascular and bronchoconstrictive diseases, two of the most common non-infectious pathologies. Patients suffering from these diseases often need to be prescribed drugs acting on β-adrenergic receptors (ADRBs): β2-agonists (usually, in lung diseases) and β1-blockers (usually, in cardiovascular pathologies).

While β_2_-agonists and β_1_-blockers are designed to target specific adrenergic receptors, they exhibit dose-dependent cross-reactivity (β_2_-agonists with β_1_-receptors and vice versa). This can lead to adverse effects in comorbid conditions (e.g., tachycardia in heart disease or bronchospasm in asthma), though selective agents mitigate these risks. Clinical guidelines recommend individualized therapy based on receptor selectivity and patient comorbidities.

Therefore, it is of considerable interest for clinical practice to assess the amounts of β1- and β2-adrenergic receptors (ADRB1 and ADRB2) in target tissues and their changes in time (dynamics) under the influence of the respective drugs.

The main targets are heart and lung tissues, which are difficult to access for laboratory analysis, except in rare cases of surgical intervention for heart disease, when a myocardial biopsy is required to clarify the diagnosis [1]. However, several studies have demonstrated a high degree of correlation between the content of ADRB in lymphocytes (or in mononuclear cells, PBMCs; here and elsewhere, PBMC refers to the total population of peripheral blood mononuclear cells before separation into subpopulations) of peripheral blood and the content of these receptors in atrial tissue [1,2,3,4,5]. For example, such a correlation was found in patients undergoing aortocoronary bypass surgery [1]. The radioligand method and the non-selective ADRB ligand cyanopindolol (CYP), which binds equally well to both ADRB1 and ADRB2, were used to assess ADRB density. It has been hypothesized that peripheral blood lymphocytes may serve as a tool to monitor changes in ADRB density in solid organs such as the heart and lungs.

It is known that ADRB1 is the most prominent subtype of adrenergic receptor in myocardial cells [6,7]. As a result of the separate determination of ADRB1 and ADRB2 content in lymphocytes versus myocardial tissue, a correlation has been established between ADRB2 densities in lymphocytes and in atrial cells (R = 0.84; *p* < 0.001) [4]; however, the correlation between ADRB2 in lymphocytes and ADRB1 in atrial cells was completely absent. The authors of this paper, as well as their colleagues [7], conclude that measuring ADRB2 content in lymphocytes is not suitable for monitoring changes in ADRB1 content in myocardium.

The quantification of ADRB1 in lymphocytes and PBMCs by radioligand assay is difficult due to the extremely low content of these receptors in these cells, which is below the sensitivity threshold of the method [8]. Nevertheless, it cannot be excluded that ADRB1 is still expressed on the cell surface of some subpopulations of PBMCs. This assumption is supported by the detection of the corresponding mRNA in PBMCs by real-time PCR [9]. The same work demonstrated a significant correlation between ADRB1mRNA in PBMCs and myocardial cells.

In the present study, we attempted to use a radioligand method previously modified by us for simultaneous quantification of ADRB1 and ADRB2 (number of molecules per cell) in different subpopulations of peripheral blood cells of healthy donors. The aim of the study is to assess the possibility of detecting ADRB1 in isolated subpopulations of PBMCs (monocytes, pan-T CD3+ cells, and NK cells), as well as to select the optimal cellular target for determining the content of ADRB2 in peripheral blood cells.

## 2. Results

The results of the simultaneous determination of ADRB1 and ADRB2 (number of molecules per cell) are shown in Figure 1 (The corresponding values of the number of ADRB1 and ADRB2 per cell and the standard measurement errors are presented in Appendix A). The figure shows that the ADRB content in all studied subpopulations of PBMCs from healthy volunteers widely varied depending on the donor.

**ADRB2** is present in all peripheral blood cell subpopulations examined. A comparison of the amount of ADRB2 present on cells from different subpopulations is summarized in Table 1, where mean values and medians are shown. It can be seen that ADRB2 is most abundant in NK cells (1334, hereinafter medians are given in parentheses), followed by PBMC (693), monocytes (453), and T cells, for which the ADRB2content was the lowest (278). We determined the Pearson correlation coefficient for the amount of ADRB2 on the cell surface for all studied cell subpopulations from healthy donors (see Table 2). A significant correlation was found between the amount of ADRB2 in PBMCs, T cells, monocytes, and NK cells (*p* < 0.05 in all cases).

**ADRB1**. The quantitation limit of our modified radioligand method is about 250 ADRB1 molecules per cell [8]. We were unable to reliably detect ADRB1 in either PBMCs or T cells, which is consistent with previously reported data for these blood cell subpopulations in healthy donors [8,10]. In the present study, we were able to reliably detect ADRB1 in the monocyte subpopulation in 43% of donors (10 of 23), and also in the NK cell subpopulation in 35% of donors (8 out of 23, with a range from 285 to 1081 molecules per cell). In some cases, the amount of ADRB1 was rather high, with more than 2000 receptor molecules per cell (see Figure 1, monocytes).It should be noted that the content of ADRB1 on the surface of monocytes does not correlate with the amount of these receptors on the surface of NK cells (*n* = 23, r = 0.27, *p* = 0.22).

## 3. Discussion

The significant variation in the average number of ADRBs per cell that we found agrees well with the data of Agapova et al. [11] who studied the binding activity of ADRB2 in peripheral blood T cells from healthy donors under strictly defined conditions. Binding activity (expressed as cpm per 10^6^ cells) was defined by the authors as the ability of the receptor to bind ^125^I-CYP in the presence of a competing ligand. As noted by the authors, who also observed a significant scatter of values, this fact does not allow us to speak of a “normal” value for ADRP2 binding activity in T cells. Whether the content of ADRB2 in blood cells of healthy donors is constitutionally determined or may vary significantly in the same donor depending on any physiologic factors remains an open question. As suggested by Krasnikova et al. [12], differences in the amount of ADRB determined in lymphocytes may depend on the level of hormones (catecholamines) in donors at the time of blood collection.

In [11,13], a subpopulation of T cells was used to monitor changes in ADRB2 binding activity under the action of the agonist salbutomol. The reasons for this choice were given as, first, that this subpopulations the most abundant and, second, that the content of ADRB2 in T cells is 2–3 times higher than in RBCs and platelets (references 9–12 from [13]). However, there were no literature data on ADRB2 content in other subpopulations of PBMC at the time of conducting and analyzing the abovementioned experiments.

According to the results shown in Table 1, ADRB2 content in the total PBMC population was higher than in the T cell subpopulation. As a significant correlation was found between the amount of ADRB2 in PBMCs, T cells, monocytes, and NK cells, it can be hypothesized that surface expression of ADRB2 is regulated in a similar manner in different subpopulations, and therefore any of them can be used to monitor changes in ADRB2 content. Obviously, it is technically easiest to manipulate with PBMCs. However, before introducing this approach into everyday clinical practice, it is necessary to conduct extended clinical studies and make sure that the response of ADRB2 to a particular drug in all subpopulations of peripheral blood demonstrates the same correlation with that in heart and lung tissues.

Although we were unable to reliably detect ADRB1 in PBMC and T cell subpopulations, it cannot be excluded that ADRB1 is present on the surface of these cells in small (below the quantitation limit of 250 receptor molecules per cell) but physiologically relevant amounts. This assumption is supported by the fact that ADRB1 mRNA was detected by real-time PCR in both PBMCs [9] and T cells [14]. The presence of ADRB1 is also confirmed by the detection of functional activity of this receptor in the study of T cells [15,16]. Specifically, monoclonal antibodies to ADRB1 stimulated in vitro proliferation of T cells obtained from the blood of patients with chronic heart failure, and also increased the production of interleukin-6 by these cells [15]. Autoantibodies to ADRB1 isolated from the blood of patients with dilated cardiomyopathy stimulated the proliferation of T cells isolated from rat blood [16].

It should be noted that to date, there are no published data concerning direct measurement of ADRB1 on the surface of human blood monocytes. The presence of ADRB1 in THP-1 cells, a monocytic cell line, was demonstrated in [8,17,18]. Treatment of THP-1 as well as primary culture of human monocytes with lipopolysaccharide and isoproterenol resulted in stimulation of interleukin-18 production by these cells [18]. And while in the case of THP-1 cells it was proved that ADRB1 is responsible for the described proinflammatory effect, the similar response of monocytes should be attributed to phenomenological observations that only indirectly indicate the presence of ADRB1 in these cells.

Our modified radioligand assay allowed us to also detect ADRB1 in NK cells. No evidence for the presence of ADRB1 in these cells could be found in the literature. Most studies on the adrenergic mechanisms of NK cells have focused on ADRB2. In [19], the adrenergic receptors of NK cells in 36 healthy donors were studied using the radioligand method. This study showed the presence of ADRB2, α1- and α2-adrenoreceptors on the surface of these cells. ADRB1 could not be detected in NK cells. The discrepancy between the results of the above work and our data can be explained by the fact that in [19], 3H-dihydroalprenolol was used as a radioactive ligand instead of ^125^I-cyanopindolol, i.e., we used a method with almost 50 times higher sensitivity [11].

The absence of correlation between ADRB1 content on monocytes and NK cells can be explained by the fact that ADRB1 expression in these subpopulations of blood cells is independently regulated. The biological meaning of this ADRB1 expression remains necessary to be elucidated in the future.

## 4. Materials and Methods

**Donors**. In a pilot, cross-sectional study, 23 healthy volunteers (29 ± 3.3 years, 13 males and 10 females) were examined. Inclusion criteria were: Healthy volunteers aged 18–40 years; BMI 18.5–30 kg/m^2^; normal ECG, blood pressure (90/60–140/90 mmHg), and spirometry (FEV_1_ ≥ 80%); no chronic diseases or medications affecting ADRB function (washout: 4 weeks for β-agonists/blockers). The exclusion criteria were pregnancy, cardiovascular/respiratory/endocrine disorders, abnormal laboratory results (glucose ≥ 6.1 mmol/L, K^+^ < 3.5 or >5.5 mmol/L), recent glucocorticoid use (≤3 months), or participation in other trials (≤3 months).

The study was conducted on the basis of FGBU “NMRCC named after Academician E. I. Chazov” of the Ministry of Health of the Russian Federation together with FGBU “Research Institute of Pulmonology” of FMBA of the Russian Federation and FGBUN “Institute of Bioorganic Chemistry named after Academicians M.M. Shemyakin and Y.A. Ovchinnikov” of the Russian Academy of Sciences.

The study was approved by the Ethical Committee of the Federal State Budgetary Institution “Research Institute of Pulmonology” of the Federal Medical and Biological Agency of Russia (protocol no. 01–21 of 14 May 2021).

All manipulations were performed in accordance with National Standard R 59778−2021 “National Standard of the Russian Federation”(https://protect.gost.ru/document.aspx?control=7&id=241615, accessed on 12 August 2025). Venous blood sampling was performed in accordance with the regulatory document ‘Procedures of venous and capillary blood sampling for laboratory research” (approved and put into effect by the Order of Rosstandart of 21 October 2021 N 1212-st).

All patients signed a written informed consent to participate in the study.

**Ligands**. The following ligands were used in this work: cyanopindolol hemifumarate (Bio-Techne Corporation, Minneapolis, MN, USA), ICI 118,551 ±)-erythro-(S*,S*)-1-[2,3-(dihydro-7-methyl-1H-inden-4-yl)oxy]-3-[(1-methylethyl)amino]-2-butanol hydrochloride (Sigma-Aldrich, St. Louis, MO, USA) and CGP 20,712 (2-((3-carbamoyl-4-hydroxy)phenoxy)ethylamino]-3-[4-(1-methyl−4-trifluoromethyl-2-imidazolyl) phenoxy]-2-propanoldihydrochloride (Bio-Techne Corporation, USA).

**Preparation of ^125^I-cyanopindolol (^125^I-CYP).** ^125^I-labeled cyanopindolol was obtained from the isotope unit of the Institute of Bioorganic Chemistry, named after Acad. M.M. Shemyakin and Y.A. Ovchinnikov Institute of Bioorganic Chemistry, Russian Academy of Sciences. The radioactive isotope of iodine as part of the Na^125^I molecule was supplied by AO Khlopin Radium Institute, St. Petersburg. The introduction of ^125^I into the cyanopindolol molecule using chloramine T was carried out using the method of Greenwood and Hunter [20] with some modifications.

For the labeling reaction, 1 μg of cyanopindolol hemifumarate and 1 mCi Na^125^I were dissolved in 50 μL of 0.2 M potassium phosphate buffer, pH 7.0. The reaction was initiated by adding 10 μL of chloramine T solution (10 mg/mL), and the mixture was incubated at room temperature for 30 s, after which the reaction was stopped by adding 10 μL of sodium thiosulfate (20 mg/mL). Then the mixture was incubated for 5 min at room temperature, and 1 μL of “cold” sodium iodide solution (6 mg/mL) was added. The target product was purified by high-performance liquid chromatography on a Diasorb 130 column (5 μm, 4 × 150 mm; Elsico, Moscow, Russia) using a Gilson ion-pair chromatography system (Villiers-le-Bel, France), with elution being performed using a linear gradient of 0–90% acetonitrile in 10% acetic acid. The fractions of ^125^I-CYP were combined, evaporated to dryness, dissolved in 70% ethyl alcohol, and stored at −20 °C.

It is known that not all ^125^I-CYP molecules may be able to bind to the receptor. The content of receptor-binding (“bindable”) fraction in ^125^I-CYP preparations was defined as the ratio of the maximum binding to ADL-7A cells expressing ADRB1 to the total amount of radioactivity in the system [11].

**The isolation of mononuclear cells and their subpopulations from human peripheral blood**. Venous blood from each donor was collected into two tubes with anticoagulant (4.5 mL per tube of 3.8% tri-sodium citrate 5,5-hydrate), 40.5 mL into each tube. Peripheral blood mononuclear cells (PBMCs) were isolated according to the Böyum method [21]. Blood was diluted with an equal volume of phosphate-buffered saline, pH 7.4 (PBS) and layered in portions of 30 mL onto 15 mL of Ficoll-1077 Diacoll solution (Dia-MLLC, Moscow, Russia) in a 50 mL tube. The tubes were centrifuged for 40 min at 400× *g* (Jouan C3i centrifuge, Thermo Fisher Scientific, Waltham, MA, USA). A ring of white cells at the border between the Ficoll and the blood sample was then taken, and the resulting PBMCs were washed three times by suspending in 50 mL of PBS and settling for 15 min at 300× *g*. The precipitated cells were suspended in 13 mL of PBS, cell counts were performed in a hemocytometer, 1 mL was taken for radioligand assay, and the remaining cells were again precipitated and suspended in PBS that contained 0.5% bovine serum albumin (BSA) and 2 mM EDTA and was preliminarily degassed and cooled to 4 °C (4 μL PBS-BSA per 106 cells). Subsequently, the suspension of PBMCs was divided into three parts as follows: 30% for the isolation of monocytes, 20% for the isolation of T cells, and 50% for the isolation of NK cells. The corresponding cell subpopulations were isolated by negative magnetic selection according to the manufacturer’s instructions using the following kits from Miltenyi Biotec GmbH (Bergisch Gladbach, Germany): “Pan T Cell Isolation Kit, human” (cat. no. 130-096-535), “Pan Monocyte Isolation Kit, human” (cat. no. 130-096-537), and “Pan NK Cell Isolation Kit, human” (cat. no. 130-092-657). The isolated subpopulations were suspended in 1 mL of PBS and cell concentration was determined by counting in a hemocytometer.

**Simultaneous determination of ADRB1 and ADRB2 content in cells of different PBMCs subpopulations and total PBMC population** was performed using a previously modified radioligand method [8]. Radioligand assay with cells of different fractions was performed on the day of their isolation. Several types of samples were prepared to determine the ADRB1 and ADRB2 content in these cells: (1) ^125^I-CYP only was added to the cell suspension. Since ^125^I-CYP is a non-selective ligand, it binds equally well to both ADRB1 and ADRB2. Some radioactivity bound to the cells in this sample is due to nonspecific binding. (2) Both ^125^I-CYP and a ADRB2-specific ligand ICI 118,551 were added to the cells, the latter being added at a concentration that would inhibit binding to ADRB2 almost completely. (3) In addition, ^125^I-CYP and ICI 118,551, an ADRB1-specific ligand CGP 20712 was added. Under these conditions, binding to ADRB2 is virtually blocked, and even the small difference between the values of bound radioactivity in samples 2 and 3 reflects the ADRB1 content in the system. We cannot use CGP 20712 alone to determine ADRB1 (in the same way as for ADRB2) as CGP 20712 ligand is not strictly specific for ADRB1, and to some extent, also binds to ADRB2. Since ADRB2 predominates in blood cells, the resulting interference prevents the separation and identification of that part of the signal difference that is due to the presence of ADRB1. (4) This sample contained, in addition to ^125^I-CYP, an excess of “cold” cyanopindolol (10 μM). The amount of cell-bound radioactivity in this sample allowed us to evaluate nonspecific binding of ^125^I-CYP. The flowchart of the method is presented in Appendix A. Primary data are contained in Appendix A.

For each of the above four samples, the measurement was performed in three parallels. The amount of ^125^I-CYP in all samples was the same and was calculated in the way that each sample contained 80,000 cpm of the “bindable” fraction. The concentration of each of the ligands ICI 118, 551, and CGP 20,712 was 0.25 μM. This concentration is saturating with respect to the corresponding receptors, and was selected by us in earlier model systems [8].

**Calculation of the content of ADRB1 and ADRB2**. ADRB content (number of molecules per cell) was calculated from the radioactivity values of cell-bound ^125^I-CYP as described in [8]. A method for calculating the numbers of ADRB1 and ADRB2 on the cell surface is also provided in the Appendix A.

**The correlation coefficient r** for the number of ADRB2 and ADRB1 on the surface of blood cells of different subpopulations, the t value and the *p* value were calculated in the Excel program using its built-in functions as described elsewhere (https://tidystat.com/calculate-correlation-coefficient-and-p-value-in-excel/ (accessed on 12 August 2025)). We used a significance level α of 0.05, that is, for all *p* values below 0.05, the correlation was considered statistically significant.

### Limitations of the Study

-The sample size is small (*n* = 23).-Our study used a homogeneous group of donors (see Section 4)—young healthy resident physicians who undergo regular medical examinations; thus, its generalization to disease states is limited.-The limitation of the method can be considered to be the large blood volume (81 mL) required to isolate cells of different subpopulations in sufficient quantity for radioligand analysis.

The use of the radioligand method may be associated with potential biases due to, inter alia, the following factors:-Receptor internalization: adrenergic receptors can undergo internalization upon ligand binding, which may lead to an underestimation of receptor numbers on the cell surface. This dynamic process can be influenced by the experimental conditions and the timing of the assay.

In our case, this factor was unlikely to have a significant effect on the estimate of the number of receptors on the cell surface because the radioactive ligand was added simultaneously together with the selective ligands (see the attached translation of Shevelev e. a, [8]).

-Pharmacological differences: radioligands may have different affinities for different subtypes of adrenergic receptors (e.g., α1, α2, β1, β2). If the ligand is not subtype-specific, the assay may not accurately reflect the density of a particular receptor subtype.

Our work utilizes the selective ligands ADRB1 and ADRB2 and takes into account the potential ability of ICI 118,551 (ADRB2-selective ligand) to partially react with ADRB1 at high concentration.

## Figures and Tables

**Figure 1 ijms-26-07894-f001:**
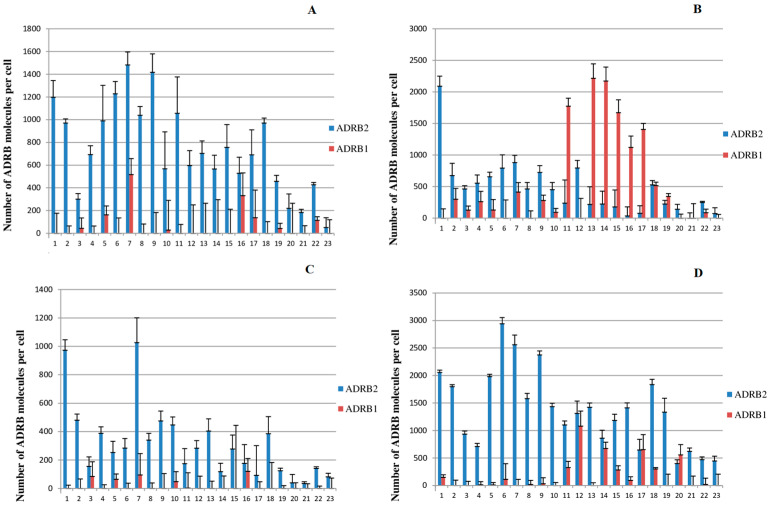
Content of ADRB1 and ADRB2 in different subpopulations of peripheral blood mononuclear cells of healthy donors. The abscissa axis denotes sequential numbers of healthy donors. (**A**), PBMC; (**B**), monocytes, (**C**), T cells, (**D**), NK cells.

**Table 1 ijms-26-07894-t001:** Mean and medians of ADRB2 content (molecules per cell) in different subpopulations of PBMCs of healthy donors, *n* = 23.

	PBMC	Monocytes	T Cells	NK Cells
MEAN	744	467	312	1366
MEDIAN	693	453	278	1334

**Table 2 ijms-26-07894-t002:** Pearson’s correlation coefficients (r) for the amount of ADRB2 on the cell surfaces of different peripheral blood subpopulations of healthy donors (*n* = 23). The *p* value was below 0.05 for all r. 95% values for CI are given in parentheses.

	Monocytes	T Cells	NK Cells
PBMC	0.62	0.73	0.85
	*p* = 0.0017	*p* = 0.00009	*p* = 0.0000002
	(0.160–0.822)	(0.230–0.878)	(0.296–0.935)
Monocytes		0.8	0.62
		*p* = 0.000004	*p* = 0.0017
		(0.270–0.912)	(0.160–0.822)
**T cells**			0.67
			*p* = 0.00047
			(0.193–0.848)

## Data Availability

All data generated or analyzed during this study are included in this published article and its Appendix A.

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
