# Peer review of "A Novel Study of β1- and β2-Adrenergic Receptors Present on PBMCs, T Cells, Monocytes, and NK Cells by Radioligand Method: Quantitation and Correlations"

_ijms, 2025, doi:10.3390/ijms26167894_

Round 1
Reviewer 1 Report
Comments and Suggestions for Authors
This work, by Peklo et al, quantitates the β1- and β2-Adrenergic Receptors across PBMCs using radioligand binding assay.
Here are my suggestions and comments –
Title
& Abstract (line 25) -
- The term PBMCs (peripheral blood mononuclear cells) consists of lymphocytes (T cells, B cells , NK cells), monocytes and dendritic cells.
It is confusing to mention T cells, Monocytes and NK cells separate from PBMC, implying they are not PBMCs.
Introduction –
- Line 27: ‘Both β2-agonists and β1-blockers are capable of interacting with both types of ADRBs, which may lead to negative side effects in cases of comorbid diseases.’
Is it implied that negative side effects are encountered only when comorbidities exist.
Also, please provide citations.
- Lines 38-40: ‘Therefore, it is of considerable interest for clinical practice to assess the amount and dynamics of β1- and β2- adrenergic receptor (ADRB1 and ADRB2) behavior in target tissues under the influence of the respective drugs.’
This is confusing- amount and dynamics of receptor behavior. May need rephrasing.
- Lines 44-45: ‘However, several studies have demonstrated a high degree of correlation between the content of ADRB in lymphocytes (or in mononuclear cells, PBMCs) of peripheral blood and the content of these receptors in atrial tissue.’
Referring to my first comment, the term PBMCs includes lymphocytes, they are not separate cells.
- It will be helpful to briefly describe the dynamics, functionality of β1- and β2- adrenergic receptors in the current context, before describing their content in PBMCs (lines 52-58).
Additionally, given this study utilizes the radioligand binding assay, a brief background on this method, and a description of why this method was chosen over others is key.
- The introduction should define the purpose of the work and its significance, including specific hypotheses being tested. Please highlight controversial and diverging hypotheses when necessary.
The introduction section is currently missing clear statements in these contexts.
Methods –
- The materials & methods sections should be moved before the results section.
- Line 165: ‘23 healthy volunteers (29 ± 3.3 years, 13 164 males and 10 females) were examined.’
What were the patients examined for?
- Lines 167-168: ‘clinical conditions that, in the opinion of the physician, interfere with participation in the study.’
It would be helpful to mention what conditions specifically led to exclusion of subjects.
- Lines 179-181: ‘Procedures of venous and capillary blood sampling for laboratory research” (approved and put into effect by the Order of Rosstandart of 21.10.2021 N 1212-st).’
Please check this sentence, it is unclear.
- Lines 233-234: ‘Simultaneous determination of ADRB1 and ADRB2 content in cells of different PBMCs subpopulations.’
Are these different PBMCs subpopulations referred to from
lines 225-226 – ‘30% for the isolation of monocytes, 20%for the isolation of T cells and 50% for the isolation of NK cells’
or
Line 221 – ‘1 mL was taken for radioligand assay’ ?
Please clarify and mention where these separate PBMC suspensions (and isolates), from lines 221 and 225-226, were utilized, so the results may be reliably related.
Results –
- It is unclear why are PBMCs being categorized separately from its subpopulation cells – T cells, monocytes, NK cells. This creates a significant error and bias in the study and statistical analysis and needs to be corrected.
PBMCs are not a separate group of cells. T cells, NK cells, monocytes are all PBMCs.
- From data in figure 1, please provide an explanation for the significantly high standard deviation for both ADRB1 & ADRB2 detection between each healthy volunteer.
- How does the binding efficiency in the assay affect the data quantitation. How was the noise in the data accounted for? It would be helpful to compute and present a subject to subject and overall signal to noise or background assessment.
- How many assay repetitions were performed per donor? This is key to provide data confidence, given the high variability across detection of not just ADRB2 but also only some patients presenting with ADRB1 expression.
Discussion –
- Line 102: ‘The significant variation in the number of ADRBs per cell that we found agrees well with the data of Agapova et al…’
This study quantitates the content of ADRB in different T cells, Monocytes and NK cells, across 23 healthy donors and not each cell.
- Lines 120-140: Again, the discussion of PBMCs separate from other subpopulations is incorrect or must be clarified.
- Lines 141-149: The discussion of ADRB1 expression in THP-1 monocytic cell line, in context to the current work is not clear.
- The limitations of the work should be highlighted.
Reviewer 2 Report
Comments and Suggestions for Authors
Peklo et al have presented the results of relevant study, which addressed the strategy to using the peripheral blood cells to monitor adrenergic receptor status. While the authors’ work possesses high potential, the manuscript may benefit from the following major and minor suggestions.
Major
- Please elaborate the introduction: add currently known studies worldwide as well and the closing of the introduction with the formation of the hypothesis and how this manuscript will answer/verify it. The current version of the manuscript does not clearly highlight the importance of the current study and its significance should be more emphasized. Please restructure it clearly stating the clinical problem, hypothesis and expected impact
- Please provide the power analysis to justify the sample size selection
- Please provide justification of the statistical analysis, including correction for multiple comparisons, confidence intervals, effect sizes.
Minor
- Missing ‘is’ in line 116. Please correct other grammatical errors along the manuscript. It will benefit to include an English-speaking colleague for the proofreading of the manuscript to improve the story flow.
- Please elaborate the discussion to be more specific about key findings, how and how they can be applied in the clinical settings.
- Please complement the discussion section by acknowledging and explaining large inter-donor variability (how it can impact future clinical applicability and possible strategies to mitigate it).
- Please provide a discussion regarding low detection rates for ADRB1 in monocytes and NK cells
- Please explain the supplementary file containing the ‘Radioligand Binding Assay for the Simultaneous Determination of β1and β2-Adrenergic Receptors in Human Blood Cells’ is it a published protocol? Do authors plan to publish it in a separate publication in international journal?
Reviewer 3 Report
Comments and Suggestions for Authors
General Comments: This manuscript presents an original study using a modified radioligand method to quantitatively assess the presence of β1- and β2-adrenergic receptors (ADRB1 and ADRB2) in subpopulations of peripheral blood mononuclear cells (PBMCs), specifically T cells, monocytes, and NK cells from healthy donors. The approach is novel, and the results provide potentially useful insights into extrapolating receptor expression from blood cells to solid organs such as the heart and lungs, which has clinical relevance in cardiovascular and pulmonary diseases. The manuscript is generally well-structured, and the methodology is clearly detailed. However, I have following comments.
Major Comments:
- While the detection of ADRB1 in NK cells and monocytes is novel, the manuscript should elaborate more clearly on the potential physiological or pathological implications of these findings. What are the hypothesized functional roles of ADRB1 in NK cells and monocytes?
- The manuscript refers to a "modified radioligand method" and provides some details, but it would benefit from a more concise schematic or flowchart (even in supplementary materials) for clarity. This would help readers follow the workflow from sample collection to receptor quantitation.
- The sensitivity limit of 250 molecules/cell is crucial. However, the authors did not address whether this detection limit is adequate to make physiologically relevant conclusions, especially when suggesting that ADRB1 may exist below this threshold.
- The discussion does not adequately address why there is no correlation between ADRB1 levels in monocytes and NK cells. More speculation or reference to possible regulatory mechanisms would improve the interpretative depth.
- The authors mention the potential for PBMCs to serve as surrogates for solid tissues in receptor monitoring. However, more detail is needed on the translational potential and limitations of this model.
- The manuscript should include a dedicated limitations section. For example:
- The sample size is small (n = 23).
- All subjects are healthy donors; thus, generalization to disease states is limited.
- The cross-sectional design precludes assessment of dynamic regulation of receptor expression.
- There is some redundancy in the introduction and discussion when explaining that ADRB1 is hard to detect in lymphocytes and that ADRB2 is used as a surrogate.

Round 2
Reviewer 2 Report
Comments and Suggestions for Authors
Peklo et al have carefully addressed my major and minor concerns and have provided reasonable justifications and clearly summaries the limitation of current study.
I would recommend proceeding with publishing this manuscript